# Polymorphism at the *CSN1S1* Locus and Energy Intake Level Affect Milk Traits and Casein Profiles in Rossa Mediterranea Goats

**DOI:** 10.3390/ani13121982

**Published:** 2023-06-14

**Authors:** Serena Tumino, Adriana Di Trana, Bernardo Valenti, Salvatore Bordonaro, Salvatore Claps, Marcella Avondo, Paola Di Gregorio

**Affiliations:** 1Dipartimento di Agricoltura, Alimentazione e Ambiente (Di3A), University of Catania, 95123 Catania, Italy; serena.tumino@unict.it (S.T.);; 2School of Agricultural, Forestry, Food and Environmental Sciences (SAFE), University of Basilicata, 85100 Potenza, Italy; 3Dipartimento di Scienze Agrarie, Alimentari ed Ambientali, University of Perugia, Borgo XX Giugno 74, 06121 Perugia, Italy; bernardo.valenti@unipg.it; 4CREA Research Centre for Animal Production and Aquaculture, Bella Muro, 85051 Bella, Italy

**Keywords:** *CSN1S1* genotype, energy intake, milk traits, casein profile, goats

## Abstract

**Simple Summary:**

The *CSN1S1* gene encodes for one of the primary milk proteins in goats. Its polymorphism strongly affects αs_1_-casein levels and plays a crucial role in determining both milk quality and quantity. The study aimed to evaluate whether a different energy intake level (70%, 100% and 150% of the total requirements indicated by INRA) could unveil any interactions with the genotype at the αs_1_-casein gene with respect to milk yield and casein profile. The results demonstrated that the higher energy input, along with the presence of the strong allele at the *CSN1S1* locus, improved milk production and casein concentrations, highlighting the existence of an interaction between α_s1_-casein polymorphisms and diet on the dairy performance of goats.

**Abstract:**

A total of twenty-seven Rossa Mediterranea lactating goats, consisting of nine homozygous for strong alleles (*AA*), twelve heterozygous (*AF*) and six homozygous for weak alleles (*FF*) at the *CSN1S1* locus, were used to evaluate the effect of genotype, diet and genotype × diet interaction on goat milk traits and casein profile. The goats were used in a 3 × 3 factorial arrangement of treatments, with three genotypes (*AA*, *AF* and *FF*) and three different energy intake levels: high (H), medium (M) and low (L). The diets supplied a complete pelleted feed containing 65% of alfalfa hay, respectively, at 150%, 100% and 70% of the total energy requirements. Milk yield was significantly affected by the genotype and diet: Lower levels were found in *FF* goats than in *AA* and *AF* genotypes (673.7 vs. 934.5 and 879.8 d/g, respectively; *p =* 0.002) as well as in goats fed with the L diet (651.5 vs. 1041 and 852.9 g/d for H and M diet, respectively, *p <* 0.001). The genotype influenced the casein profile. Specifically, *AA* goat milk exhibited higher concentrations of total casein and α_s1_-casein compared to *AF* and *FF* genotypes (for total casein and αs1-casein, respectively: 24.9 vs. 20.4 and 19.8 g/kg, *p =* 0.001; 7.2 vs. 3.7 and 0.7 g/kg, *p <* 0.001), while the *FF* genotype showed higher values for α_s2_-casein concentrations compared to homozygous *AA* and heterozygous *AF* goats (3.1 vs. 2.4 and 2.5 g/kg, respectively, *p <* 0.001). A significant genotype x diet interaction occurred for αs2-casein levels (g/kg) (*p =* 0.034) and αs_1_-casein yields (*p =* 0.027): The αs_2_-casein level was not affected by the diet in *AA* goats, whereas it increased with energy intake in *AF* and *FF* genotypes. Conversely, the αs_1_-casein yield gradually increased with energy intake in *AA* and *AF* groups, whereas the diet in *FF* goats did not modify it. The results demonstrated that high energy input, as well as the strong allele at the *CSN1S1* locus, enhanced milk production and casein concentrations. Furthermore, they confirmed the existence of an interaction between α_s1_-casein polymorphism and diets, influencing the milk casein composition and yield.

## 1. Introduction

The *CSN1S1* gene is responsible for encoding one of the primary milk proteins in goats and plays a crucial role in determining milk quality. Genetic polymorphism in this gene can have a significant impact on milk-related traits, including milk yield, composition, fat concentration, fatty acid composition and clotting properties [1,2,3,4,5]. Eighteen alleles of the *CSN1S1* gene have been identified in goats and are categorized according to their effects on αs_1_-casein synthesis in milk. Strong alleles (A, A3, B1, B2, B3, B4, C, H, L and M) each produce 3.6 g/L of αs_1_-casein; intermediate alleles (E and I) produce 1.1–1.6 g/L; weak alleles (F, D and G) produce 0.45–0.6 g/L; null alleles (01, 02 and N) are associated with the absence of αs_1_-casein in milk. There are different frequencies of *CSN1S1* alleles among Italian goat breeds. A, E and F alleles are the most common in northern Italian breeds, while strong and F alleles prevail in the autochthonous goat population in Southern Italy, whereas the most abundant *CSN1S1* allele observed in Sarda goats is B. Conversely, C, H, N and 01 alleles have a low frequency in Italian goat breeds [6]. In the Mediterranean environment, variations in feeding levels are common in extensive goat breeding systems due to the variability in the availability of forage resources linked to climate change. Such variations in nutrient availability have significant effects on the conversion efficiency of the diet into milk. Furthermore, the feeding level can also impact the qualitative characteristics of milk due to changes in the regulation of the expression of genes involved in milk component synthesis. However, the effect of feed restriction is not unique because it probably depends on its duration and intensity or the lactation stage in which it occurs, as highlighted in a bibliographic review by Leduc et al. [7]. Few studies have been conducted on the interference of the diet on milk yield or qualitative characteristics of milk based on the α_s1_-casein genotype. The effects of diet energy levels [1], fresh forage [2], extruded linseed [8] and dietary protein content [9] were investigated in goats that are genetically predisposed to produce more (strong) or less (weak) α_s1_-casein, demonstrating significant effects of the interaction between diet and the α_s1_-casein genotype on some milk traits. In a previous study. Pagano et al. [1] fed ad libitum Girgentana goats using diets with different energy contents (modulating the hay content in the diet from 35 to 100%) and observed that the higher energy input improved the efficiency of the diet-to-milk transformation and casein yields in *AA* goats. In contrast, it did not exert noticeable effects in *FF* goats. However, in those experimental conditions, extremely high intake was recorded for the three diets, resulting in very high energy inputs even when the diet consisted of 100% hay. Therefore, the objective of the present experiment was to evaluate whether different energy intake levels (70%, 100% and 150% of the total requirements indicated by INRA) [10] could unveil any interactions with the αs_1_-casein genotype with respect to milk traits and casein profiles.

## 2. Materials and Methods

The trial was carried out in the CRA-ZOE farm (Research Unit for Extensive Zootechnics, Via Appia, Bella Scalo 85054—Muro Lucano) located at 360 m a.s.l. (40°21′ N; 15°30′ E).

Twenty-seven multiparous Rossa Mediterranea goats, homogeneous in terms of days of lactation (49 ± 5 d), were used in the study. At the beginning of the test, the animals had a body weight of 49.1 ± 1.2 kg and milk production of 1.3 ± 0.3 kg/d. The subjects were selected from a large herd based on their genotype at the *CSN1S1* locus:Group *AA*: 9 goats homozygous for strong alleles at the αs_1_-casein locus;Group *AF*: 12 goats heterozygous for alleles at the αs_1_-casein locus;Group *FF*: 6 goats homozygous for weak alleles at the αs_1_-casein locus.

All goats used in the experimental trial were characterized by *CSN1S2^A^* and *CSN2^A^* alleles associated with normal amounts of αs_2_-casein and β-casein contents, respectively. Goat DNA was obtained from hair bulbs following the method described by Bowling et al. [11]. The animals’ genotypes were determined using PCR methodology, as suggested by Jansà Pérez et al. [12], Ramunno et al. [13], Ramunno et al. [14] and Cosenza et al. [15].

### 2.1. Feeding Treatment

All animals were housed individually in single pens and subjected to three feeding treatments. The diet provided for the goats consisted of a single pelleted feed with a diameter of 6 mm and contained 150 g of mixed-hay dry matter. The constituent ingredients of the pellet and its chemical composition are detailed in Table 1.

The three feeding treatments were as follows:H diet: Characterized by an energy intake equal to 150% of energy requirements;M diet: Characterized by an energy intake equal to 100% of energy requirements;L diet: Characterized by an energy intake equal to 70% of energy requirements.

The energy requirements were calculated according to the INRA System [10], and they were determined as follows:−Net energy requirements for maintenance (UFL/d) = 0.01 × live weight (kg) + 0.19;−Net energy requirement for milk production (UFL/d) = milk yield normalized at 3.5% of fat, kg/d × [0.4 + 0.0075 × (fat, g/kg − 35)].

The animals were divided into 3 blocks consisting of 3 *AA*, 4 *AF* and 2 *FF* goats. Following a 3 × 3 Latin square scheme, the goats received the three different diets (H, M and L) in succession in three phases (1, 2 and 3). Each phase consisted of 10 days of adaptation to the scheduled feeding treatment and an 8-day experimental period in which sample measurements were performed. The individual feeding intake was envisaged by weighing the provided feed and measuring the amount refused. Nevertheless, all administered feed was always consumed by the animals. The experiment lasted for a total of 54 days.

### 2.2. Milk Production and Samples Collection

Individual milk production and milk samples were collected from morning and evening milking at d 5 and 8 of each 8 d collection period. Then, proportional volumes between the milk amount recorded at the respective times of morning and evening milking were pooled to obtain the individual final samples intended for analyses. The collected milk samples were stored at 4 °C until chemical analysis was conducted.

### 2.3. Milk Analysis

Milk samples were analyzed for lactose, fat, protein and urea by using an infrared method (Combi-foss 6000, Foss Electric, Hillerød, Denmark).

### 2.4. Capillary Zone Electrophoresis (CZE)

A Beckman P/ACEMDQ Capillary Electrophoresis system controlled by 32 Karat Software, version 8.0 (Beckman Instruments, Fullerton, CA, USA), and equipped with a UV detector set at 214 nm was used in this study. Separations were carried out using an uncoated fused silica capillary (57 cm length, 50 m i.d., 375 m OD slit opening 100 × 800 m; Beckman Instruments, Fullerton, CA, USA). Sample solutions were injected for 20 s at 0.5 psi. Electrophoresis runs were carried out at 45 °C with a linear voltage gradient from 0 to 25 kV in 3 min, followed by a constant voltage at 25 kV. Buffers for CZE analyses were prepared according to Heck et al. [16]. The sample buffer (pH 8.6 ± 0.1) comprised 167 mM hydroxymethyl-aminomethane (TRIS—BIO-RAD, Hercules, CA, USA), 42 mM 3-morpholinopropane sulphonic acid (MOPS—SIGMA, Burlington, MA, USA), 67 mM ethylenediamine–tetraacetic acid disodium salt dihydrate (EDTA—SIGMA), 17 mM _D-L_dithiothreitol (DTT—BIO-RAD), 6 M urea (BIO-RAD) and 0.05% (*w*/*w*) hydroxypropyl methylcellulose (HPMC—SIGMA). The running buffer (pH 3.0 ± 0.1) comprised 0.19 M citric acid (CARLO ERBA, Milan, Italy), 20 mM sodium citrate (CARLO ERBA), 6 M urea and 0.05% (*w*/*w*) HPMC. Individual samples were prepared by mixing individual milk and sample buffers (1:1.5); after 1 h at room temperature in the dark, samples were centrifuged at 5000× *g* for 5 min, and the top fat layer was removed. Samples were analyzed without further preparation. The caseins were identified by reference to the literature [17,18,19,20]. Since CZE peak areas are inversely correlated with migration velocity, the relative concentration of individual proteins was determined based on the corrected area using Equation (1), as reported by Heck et al. [16]:(1)Cx=Axtx∑i=1n(Aiti)∗100%
where *C_x_* is the relative concentration, *A_x_* is the area in the electropherogram, *t_x_* is the migration time of protein *x* and *n* is the total number of peaks that together comprise 100% of the area. The quantities of individual caseins were calculated from the total casein amount, as reported by Valenti et al. [21].

### 2.5. Statistical Analysis

Individual milk yield and composition data were analyzed using the GLM procedure for repeated measures (SPSS for Windows, Inc., Chicago, IL, USA). The model included the αs_1_-casein genotype, energy intake levels, blocks, periods, and αs_1_-casein genotype x energy intake levels. Pre-experimental milk production and composition were used as covariates for milk production and gross composition, respectively. When the covariate was not significant, it was removed from the model. Individual data for casein profile and body weight were analyzed using the GLM univariate procedure (SPSS for Windows, Inc., Chicago, IL, USA), and analyses included the main effect of the αs_1_-casein genotype, energy intake levels, blocks and genotype × diet interactions. Differences between means were tested using least significant differences (LSDs).

## 3. Results

Table 2 reports milk yield and the gross composition of milk. Milk yield was significantly affected by the genotype and energy intake; mean daily production was significantly lower in *FF* compared to *AF* and *AA* goats and increased linearly with increasing energy levels. Fat and protein were significantly higher, and lactose was lower in *AA* goats compared to the other genotypes. Increasing energy levels significantly reduced the fat percentage but did not influence protein or lactose levels. No treatment effects were evident for urea. For each parameter, no interaction between these genotypes and diet was evident.

In Table 3, the casein profile and yield are reported. The total casein level (g/kg) was higher in *AA* compared to the other genetic groups, whereas no difference was evident between *AF* and *FF* goats. The αs_1_-casein linearly decreased from strong homozygous to weak homozygous goats. A significant opposite trend was evident for αs_2_-casein, increasing from *AA* to *FF*. β-casein and k-casein were not affected by the genotype. The effect of the energy level was evident only for α-s_2_ casein, which was significantly lower in the L diet compared to M and H diets.

Regarding casein yields (g/d), the total casein and αs_1_-casein decreased from *AA* to *FF* goats. Β-casein and k-casein were lower in the *FF* group, whereas no differences were observed between *AA* and *AF* genotypes. The casein yield was also affected by feeding: Total caseins, αs_2_-casein, β-casein and k-casein regularly increased from the L to H diet. αs_1_-casein was significantly higher in the H group, whereas no difference was observed between L and M diets.

A significant genotype x diet interaction was found for αs_2_-casein levels (g/kg) and αs_1_-casein yields (g/d) (Figure 1). The αs_2_-casein level was not affected by diet in *AA* goats, whereas it increased with energy intake level in *AF* and *FF* genotypes. On the contrary, the αs_1_-casein yield gradually increased as energy input increased in *AA* and *AF* groups, whereas it was not modified by the diet in *FF* goats.

## 4. Discussion

The pelleted diets individually offered to the animals (1.04, 1.49 and 2.23 kg DM/d, respectively for L, M and H diets) and the hay were always entirely consumed; therefore, the energy input coincided with the expected input, equal to 70%, 100% and 150% of total needs. Including 65% pelleted hay ensured an adequate supply of neutral detergent insoluble (NDF). The goats showed no clinical signs of metabolic disorders during the trial. On average, milk yield was lower in the weak αs_1_-casein genotype. Similar results were previously found in Girgentana goats [1,5,22], whereas opposite results were found by Chilliard et al. [8] in Alpine goats. Fat and protein were higher in *AA* goats, whereas lactose was lower. No parameter showed significant differences between *AF* and *FF* goats. As expected, the protein content was positively associated with the higher capability of αs_1_-casein synthesis in goats with strong alleles. Higher levels of fat in goats carrying strong alleles were reported in previous studies [1,8,9,23]. In a previous research study, Ollier et al. [24] suggested a downregulation of the expression of genes involved in milk fat synthesis in goats with weak alleles at the *CSN1S1* locus, which could partially explain our results. Milk urea was not affected by the genotype. In a previous paper, Avondo et al. [5,22] found lower urea contents in milk from goats carrying strong alleles at the αs_1_ locus, highlighting this expected result. The concentration of urea in milk is related to the degree of the efficiency of utilization of the dietary protein by rumen microorganisms due to the dietary energy-to-protein ratio. Usually, milk urea increases as the energy-to-protein ratio of the diet decreases. It should be underlined that in the cited studies, animals were fed diets containing proteins exhibiting different degradability [5] or freely selected diets containing different protein levels and degradability [22]. Conversely, in the present paper, the animals always received the same pelleted feed with a constant energy-to-protein ratio and protein quality across different feeding treatments. This could explain the lack of differences between groups in our experimental conditions.

As expected, with varying energy intake, milk yield increased, and fat decreased (due to a dilution effect), whereas no effects of the diet on protein, lactose, or urea were evident.

Live weight was significantly higher in *FF* compared to *AF* and *AA* goats. It is likely that lower production in *FF* goats resulted in fattening, suggesting that energy input in this genotype was channeled relative to the accumulation of fat reserves rather than milk production. Indeed, the energy conversion efficiency into milk was significantly worse in this genetic group compared to *AF* and *AA*. Nevertheless, body condition score (BCS) values did not show substantial differences between genotypes. The feed restriction at 70% of energy requirements (L diet) resulted in a corresponding drop in production equal to 76% compared to the 100% diet, while the increased energy intake to 150% of the basic diet (diet H) resulted in an increase in milk production by only 122%. These results were confirmed by the significant worsening of the energy transformation efficiency in milk with the highest energy input. No interaction effect between the experimental factors was reported for milk yield and gross composition. The higher milk production, reported by Bonanno et al. [3] and Pagano et al. [13] for Girgentana goats with the strong genotype and that were fed with high-energy diets, was not found in our experimental conditions. It could be hypothesized that the Rossa Mediterranea breed may have responded differently to variations in energy intake compared to the Girgentana breed used in previous studies. A different response between breeds to feed restriction was recently observed in other species. In a recent study that aimed to investigate the effects of feed restriction on mammary miRNAs and coding gene expression in mid-lactation cows, feed restriction modified the expression of 27 miRNAs and 374 mRNAs in mammary glands from Holstein cows, whereas no significant miRNA change was observed in Montbeliarde cows [25].

Regarding the casein profile, all goats used in the trial were characterized by *CSN1S2*^A^ *CSN2^A^* alleles, which are associated with normal levels of αs_2_-casein and β-casein content. Furthermore, it should be noted that the tested population was monomorphic at the k-casein locus, as already observed by Albenzio et al. [26] in Garganica goats. Therefore, assuming that other conditions are equal, the difference in αs_1_-casein levels observed between genotypes was coherent with the capability of casein synthesis for the *A* and *F* alleles, and these are indicated as equal to 3.6 and 0.45 g/L [27]. As_2_-casein, although monomorphic in our experimental test, showed a gradual increase from the *CSN1S1 ^AA^* to the *CSN1S1 ^FF^* group, thus highlighting an opposite trend compared to αs_1_-casein. The hypothesis that αs_1_-casein in *FF* goats could be partially compensated by the synthesis of other caseins has already been advanced by Valenti et al. [21]. However, despite the trend of αs_2_-casein, the total casein gradually increased from the *FF* genotype to the *AA* genotype. Beyond the results between genetic groups, it should be noted that significant correlations were found between casein fractions. In particular, αs_2_-casein was negatively correlated with αs_1_-casein (r = 0.36) and positively correlated with β-casein and k-casein (r = 0.68 and 0.53, respectively). On the contrary, Song et al. [28] found that *CSN1S1* overexpression markedly downregulated β-casein expression but had no significant effect on the expression level of αs_2_-casein and κ-casein.

Casein yields, of course, were associated with the daily milk yield, thus resulting in gradual and significant reductions in total casein and αs_1_-casein production from the *AA* genotype to the *FF* genotype, and it was significantly lower in *FF* goats compared to *AA* and *AF* goats for β-casein and k-casein. The production of αs_2_, having shown an opposite trend relative to αs_1_ in the casein profile, compensated for the differences in milk production between genotypes, making its production non-significantly affected by genotypes.

The energy input only significantly modified the levels of αs_2_-casein, which were significantly lower in the L diet than in the M and H diets but strongly reduced all casein yields as the energy level decreased. Ollier et al. [29], by studying the impact of 48 h food deprivation on goat mammary gene expression, found a downregulation for αs_1_ and αs_2_ precursors. Tsiplakou et al. [30], who investigated the impact of long-term underfeeding and overfeeding on the expression of six major milk protein genes (αs_1_-casein, αs_2_-casein, β-casein, κ-casein, α-lactalbumin and β-lactoglobulin) in the mammary goat tissue, observed that underfeeding led to a decrease in the expression level of all genes, suggesting that our experimental L diet could negatively impact milk protein synthesis.

A significant interaction genotype x diet was found for αs_2_-casein levels and αs_1_-casein yields, which showed different sensitivity to the energy intake level in different genotypes: αs_2_-casein levels gradually increased as a function of the energy input only in *AF* and *FF* genotypes. In contrast, αs_1_-casein yields gradually increased from group L to group H only in *AF* and *AA* genotypes. Both results substantially confirm previous findings reported by Valenti et al. [21] in Girgentana goats fed ad libitum with diets at different energy content. In particular, the maximum difference in αs_1_-casein synthesis between strong and weak genotypes, equal to 8.18 g/d, occurred when the animals received the highest energy level. Conversely, the lowest difference (4.13 g/d) was reached when they were fed the lowest energy level. Schmidely et al. [31] found that the difference in milk protein content between *AA* and *FF* goats was higher when the balance was positive. The authors suggested that the maximal difference between the two genetic variants was related to the energy status of the goat. In those experimental conditions, the effect of the genotype on milk production was not evidenced. In our conditions, the interaction between the genotype and the αs_1_-casein yield was not evident in the concentration of αs_1_-casein in milk (g/kg), and this is likely due to a dilution effect caused by varying milk production, masking the interactions’ impact.

## 5. Conclusions

In the presence of the strong allele at the *CSN1S1* locus, the milk yield and percentages of fat, protein and αs_1_-casein were higher, whereas the αs_2_-casein percentage was lower. The hypothesis that a different energy intake level could interfere with milk traits and the casein profile of goats with different *CSN1S1* genotypes was partially confirmed. The results demonstrated that the αs_1_-casein yield increased with energy input in goats carrying the strong allele, whereas no difference was evident in homozygous goats carrying the weak allele. This confirms the existence of an interaction between *CSN1S1* polymorphism and the diet, which influences milk casein composition and yield. These results suggest the possibility of adapting the diet to the genotype in order to improve its transformation efficiency.

## Figures and Tables

**Figure 1 animals-13-01982-f001:**
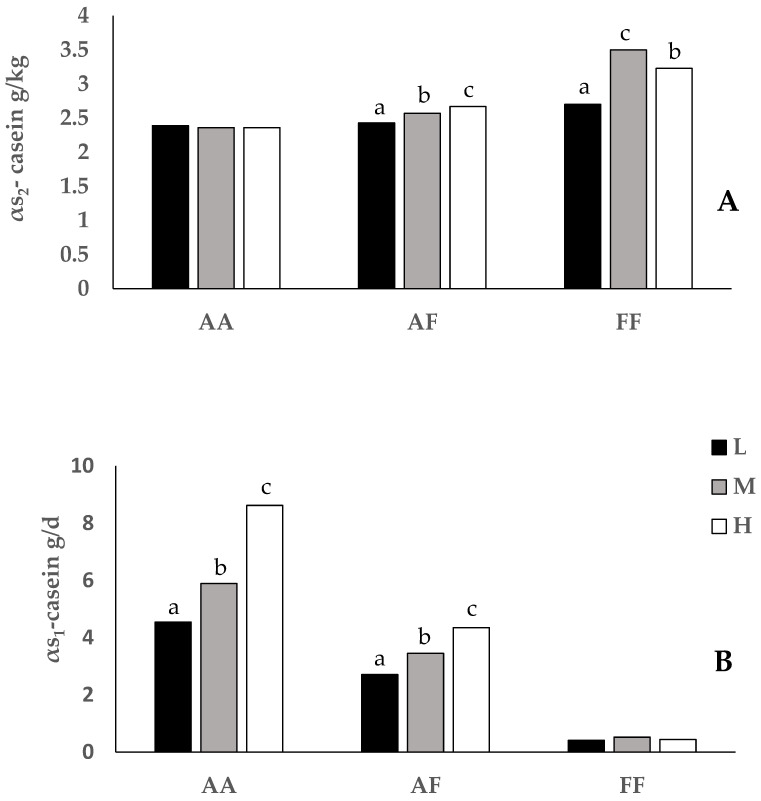
Interaction between the *CSN1S1* genotype (*AA*, *AF* and *FF*) and diet (L, M and H) for the αs_2_-casein level (**A**) and **α**s_1_-casein yield (**B**). Values within genotypes with different superscript letters are significantly different (*p <* 0.05).

**Table 1 animals-13-01982-t001:** Diet components and chemical composition.

** *Ingredients* **	**% As Fed**
Pelleted alfalfa hay	65.0
Maize grain	15.8
Barley grain	8.2
Soybean meal	3.0
Carob pulp	3.0
Maize gluten meal	3.0
Vitamin and mineral premix	2.0
** *Chemical Composition* **	
Dry matter (DM) %	85.7
Crude protein % DM	15.2
Crude fiber % DM	23.1
Neutral detergent insoluble (NDF) % DM	44.5
Ether extract % DM	2.6
Ash % DM	10.6
NFC % DM	27.1
Starch % DM	19.6
UFL	0.82

**Table 2 animals-13-01982-t002:** Milk yield and gross composition of milk.

	*CSN1S1* Genotype (G)			Energy Intake Level (E)			Significance (*p*)			SEM
	*AA*	*AF*	*FF*	L	M	H	G	E	G × E	
Milk yield g/d	879.8 ^b^	934.5 ^b^	673.6 ^a^	651.5 ^a^	852.9 ^b^	1041.0 ^c^	0.002	<0.001	0.547	34.0
Fat %	5.24 ^b^	4.16 ^a^	3.80 ^a^	5.68 ^c^	4.30 ^b^	3.61 ^a^	<0.001	<0.001	0.590	0.14
Protein %	4.77 ^b^	3.92 ^a^	4.08 ^a^	4.43	4.29	4.14	<0.001	0.975	0.974	0.07
Lactose %	4.38 ^a^	4.62 ^b^	4.65 ^b^	4.49	4.50	4.62	0.010	0.695	0.968	0.03
Urea mg/dL	481.6	502.6	517.7	481.1	510.3	503.2	0.311	0.992	0.650	8.78
Live weight kg	40.8 ^a^	39.4 ^a^	44.1 ^b^	39.3	41.7	42.3	0.017	0.114	0.982	0.51
Body condition score (BCS)	2.49	2.67	2.70	2.60	2.63	2.60	0.141	0.993	0.995	0.03
Milk efficiency ^1^	1.53 ^a^	1.42 ^a^	1.96 ^b^	1.35 ^a^	1.54 ^a^	1.90 ^b^	<0.001	<0.001	0.408	0.05

^1^ UFL intake/kg latte. ^a,b,c^ Values within a row without a common superscript letter are significantly different (*p <* 0.05).

**Table 3 animals-13-01982-t003:** Caseins profile and yield.

	*CSN1S1* Genotype (G)			Energy Intake Level (E)			Significance (*p*)			SEM
	*AA*	*AF*	*FF*	L	M	H	G	E	G × E	
Caseins profile g/kg milk										
Total casein	24.9 ^b^	20.4 ^a^	19.8 ^a^	21.2	22.3	21.4	0.001	0.523	0.260	0.54
αs_1_-casein	7.22 ^c^	3.70 ^b^	0.67 ^a^	4.36	3.83	3.74	<0.001	0.873	0.910	0.32
αs_2_-casein	2.38 ^a^	2.55 ^b^	3.14 ^c^	2.48 ^a^	2.77 ^b^	2.74 ^b^	<0.001	0.023	0.034	0.05
β-casein	12.7	11.8	13.4	12.1	13.1	12.4	0.052	0.511	0.090	0.29
k-casein	2.53	2.29	2.59	2.31	2.61	2.45	0.164	0.285	0.421	0.06
Casein yield g/d										
Total casein	20.9 ^c^	19.4 ^b^	13.3 ^a^	13.8 ^a^	18.4 ^b^	23.0 ^c^	0.001	0.001	0.411	0.97
αs_1_-casein	6.09 ^c^	3.50 ^b^	0.46 ^a^	2.82 ^a^	3.31 ^a^	4.45 ^b^	<0.001	0.006	0.027	0.36
αs_2_-casein	2.06	2.44	2.12	1.63 ^a^	2.30 ^b^	2.86 ^c^	0.072	0.002	0.806	0.10
β-casein	10.6 ^b^	11.3 ^b^	8.94 ^a^	7.88 ^a^	10.7 ^b^	13.1 ^c^	0.046	0.004	0.553	0.51
k-casein	2.16 ^b^	2.15 ^b^	1.74 ^a^	1.51 ^a^	2.12 ^b^	2.58 ^c^	0.033	0.001	0.475	0.10

^a,b,c^ Values within a row without a common superscript letter are significantly different (*p <* 0.05).

## Data Availability

The datasets used and analyzed during the current study are available from the corresponding author upon reasonable request.

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
