# Peer review of "Polymorphism at the CSN1S1 Locus and Energy Intake Level Affect Milk Traits and Casein Profiles in Rossa Mediterranea Goats"

_animals, 2023, doi:10.3390/ani13121982_

Round 1

Reviewer 1 Report

1. There are many mistakes, non-standard and inconsistent writing in the original manuscript. Such as “P=0.002” in the Line 33 and “P<0.001” the in Line 35, “1041,0” in the Line 34, “CSN1S1 alleles” in the Line 57 and “CSN1S1 allele” in the Line 60, “(40°21'N; 15°30'E.” in the Line 91, “0.70%” in the Line 85…

2. What is difference between CSN1S1 gene and CSN1S1 locus? Author sometimes use CSN1S1 locus and the word “CSN1S1’’ is not italicized.

3. In introduction, author needs to provide a detailed description of the background of the gene, the relationship between different genotypes, different alleles, and phenotype.

4. In discussion, author still needs to thoroughly revise the logical of the discussion. Why there are differences among the three diets of the AA genotype, while there are no differences among the three diets of the FF genotype. What is the interaction relationship?

5. I think the major problems are in Materials and Methods. In my opinion, background of animals is not enough, sample size is too small to support the conclusions obtained from the experiment.

Extensive editing of English language required

Author Response

Please, find our replies in the attached file

Reviewer 2 Report

This paper describes the effect of energy intake on milk casein composition in goats displaying different genotypes at the CSN1S1 locus. The effect of CSN1S1 genotype on casein composition in milk (especially the compensation of as2 casein in FF goats) and the effect of feeding strategy on casein production from goats with different CNS1S1 genotypes have been reported for a long time. Morever, a very quite similar study has been published by Veneti et al in 2012 on Girgentana goats. Thus, this work is not very original, since it is the same work on a different breed. Discrepancies of results between the 2 studies (in 2 different breeds) are not underpinned. Why? What mechanims? The only new result is the genotype x diet interaction for as2casein level, a quite little poor stuff for writing a paper about.

Lines 99-100: aS2: what does “normal alleles” mean?? As2 can display 3 alleles: A B or C. Normal means A alleles only?

Feeding treatment: how do you measure real feeding intake? Did you weight offered/refused feed? Not precised in the paper

Lines 13-132: Milk samples collection (§2.2): what does that mean: “proportional volumes of morning and evening milk were gathered? You mixed equal volumes of morning and evening milk?

I do not really understand table 2. For each group (L, M and H energy intake level), you measure all milk parameters, whatever the genotype is (since each group have mixed aS1 casein genotypes)? Please rewrite table 2 considering genotype for each energy intake level (as done for Girgentana goats in 2012)

Minor:

Line 19 and line 85: 0.70%?

Line 29: intake

Author Response

Please find our replies in the attached file

Reviewer 3 Report

The reviewed manuscript is an original development of the well-known issue of casein gene polymorphism in goats. The authors have extensive experience in this area, as evidenced by previous publications in top-ranked scientific journals. All comments to the work are included directly in the text, they have the nature of editorial corrections, as the work was prepared very carefully, both in terms of methodology and the form of presenting the results. I sugget to publish it, as a minor revision.

Author Response

(The authors gave the same response as above.)

Round 2

Reviewer 1 Report

the original manuscript has been greatly improved through revision. However, there are still many minor errors in writing.

There are still many minor errors in writing.

Author Response

We wish to thank the reviewers for their time and for the criticism aimed at improving the quality of the manuscript, we hope to be able to satisfy their suggestions with this review.

we have provided the minor revision to the English form requested by the referee 1